# Health Programmes for Older Adults Who Are the Primary Family Caregivers for Their Partners: A Scoping Review

**DOI:** 10.3390/healthcare12242523

**Published:** 2024-12-13

**Authors:** Anabel Chica-Pérez, Lucía Martínez-Sola, Matías Correa-Casado, Cayetano Fernández-Sola, Karim El Marbouhe El Faqyr, José Manuel Hernández-Padilla

**Affiliations:** 1Emera Elderly Care Home, 04007 Almeria, Spain; acp819@inlumine.ual.es; 2Department of Nursing, Physiotherapy and Medicine, University of Almeria, 04120 Almeria, Spain; lms740@inlumine.ual.es (L.M.-S.); cfernan@ual.es (C.F.-S.); eek918@ual.es (K.E.M.E.F.); jhp861@ual.es (J.M.H.-P.); 3Faculty of Health Sciences, Universidad Autónoma de Chile, Santiago 7500000, Chile

**Keywords:** aged, caregivers, health promotion, partner, programmes

## Abstract

**Aim:** To examine and map health programmes that have been implemented and evaluated to improve health outcomes amongst older adults who are the primary family caregivers for their partners. **Methods:** A scoping review was carried out, following the methodology of the Joanna Briggs Institute (JBI) and PRISMA-ScR reporting guidelines. The search was conducted in six databases (PubMed/Medline, Cochrane, CINAHL, Web of Science, Scopus, and EMBASE) between December 2023 and March 2024. **Results:** Eleven studies were included, covering different health programmes implemented and evaluated with the aim of improving health outcomes in older adults who assume the role of primary family caregiver for their partner. The programmes were carried out by multidisciplinary teams and psychologists. The programmes varied in content, frequency, and duration. Discrepancies were found in the effects of the interventions on stress, depression, quality of life, and caregiver burden. **Conclusions:** This scoping review shows that programmes aimed at improving health outcomes in older adults who are the primary family caregiver for their partner vary widely in both content and effectiveness. While some interventions reduced the burden and psychological distress, others did not show clear improvements in quality of life. It can be concluded that there is a need for randomised controlled trials that rigorously evaluate the outcomes of long-term, personalised interventions.

## 1. Introduction

Adopting a family caregiving role has an impact on the biopsychosocial health of older adults [1]. A family caregiver is defined as a person who provides unpaid care to a relative related by blood or marriage [2]. It is estimated that between 5 and 21% of the global population are family caregivers [3], while the available data indicate that 13% of the population in Spain assumes this role [4]. In the case of dependent older adults, the available evidence suggests that it is primarily their partners who assume the role of primary family caregiver [1,5,6,7,8,9,10]. This duty is known to be a risk factor for the development of numerous health problems [11] and is associated with fatigue and burnout [12]. In addition, the health status of family caregivers is further affected in the case of older adults [13], who perceive accessing health services as a frustrating and exhausting process [14,15]. In this context, it is important for health professionals to design, implement, and evaluate the effects of specific interventions to improve the health and support offered to older adults who take on the role of primary family caregiver for their partners [6]. 

In general, family caregiving is associated with higher levels of depression [16] and poorer quality of life [17]. In the case of older adults, this role is linked to an increased likelihood of experiencing pain [18], greater mental health impairment [19], and reduced quality of life [20]. Anxiety, depression, and caregiver burden are the main factors contributing to the deterioration of older adult family caregivers’ quality of life [17]. Moreover, the increased risk of hospitalisation among older adults [21] is an added stress when they assume the role of primary family caregiver [22]. Furthermore, the scarcity of support services for older adult primary family caregivers, as well as the difficulties they face in navigating the health care system [14], may lead them to seek help less frequently [23,24].

The available evidence suggests that numerous interventions have been implemented and evaluated worldwide with the aim of improving the biopsychosocial health of family caregivers [8,25,26,27]. However, most of these interventions were not designed with the caregiver’s age in mind, which could be a determining factor in their health status [28]. Although some authors suggest that specific health programmes could improve the quality of life of older adult family caregivers [29], no literature reviews have been found to assess the scope of these programmes. The general objective of this scoping review was to examine and map health programmes that have been implemented and evaluated with the aim of improving the health of older adults who assume the role of primary family caregiver for their partner. The specific objectives of this review were: (1) to identify the health programmes designed for older adults who are primary family caregivers for their partners and assess their evaluation outcomes, (2) to describe the key characteristics of the programmes, including their duration and frequency, (3) to identify the main health condition of the people receiving informal care, (4) to explore the outcome domains addressed by the interventions included in the review, and (5) to evaluate the specific outcomes that have been measured in the studies assessing the health programmes included in the review.

## 2. Materials and Methods

### 2.1. Design

This scoping review follows the Joanna Briggs Institute (JBI) methodology for scoping reviews [30] and complies with the Checklist for Systematic Reviews and Meta-Analyses for Scoping Reviews (PRISMA-ScR) [31].

The protocol of this study was registered with INPLASY (Registration Number: INPLASY2024120037, DOI: https://doi.org/10.37766/inplasy2024.12.0037).

### 2.2. Search Strategy

To select the studies, a comprehensive search was carried out between December 2023 and March 2024 using a three-phase process. In the first phase, an initial search was conducted in the following databases: MEDLINE (via PubMed), Cochrane, EMBASE, CINAHL, Web of Science, and SCOPUS. During this phase, keywords from the titles and abstracts of the main articles were identified, including terms like “elderly”, “older adults”, “caregiver”, “health promotion”, and variations related to these topics. In the second phase, the natural language found in the first phase was combined with MeSH terms related to the topics, such as “aged”, “caregiver”, and “health promotion”. For this purpose, the Boolean operators “OR” and “AND” were used, and a comprehensive search strategy was designed. Finally, in the third phase, the search strategy was adapted for each database. The combination of natural language and MeSH keywords resulted in the following search strategy used for PubMed: ((elderly) OR (older adults) OR (aged) OR (older) OR (olders) OR (aged)) AND ((care-giver) OR (caregivers) OR (carer) OR (caregivers)) AND ((spousal) OR (spouse) OR (partner) OR (couple)) AND ((intervention) OR (interventions) OR (program*) OR (programme) OR (health programme) OR (health programme) OR (health promotion)). This search strategy was used as a guide for the rest of the databases with the following modifications (see Appendix A).

### 2.3. Eligibility Criteria

In line with the JBI methodology for scoping reviews [30], the PCC methodology (P: participants; C: concept; C: context) was applied to define the inclusion criteria:Participants: studies focusing on older adults with a mean age of 65 years or above who undertake the role of primary caregiver for their partner.Concept: studies involving programmes/interventions implemented and evaluated to improve their health.Context: studies conducted in community settings.

This review included quantitative, qualitative, and mixed methods studies. In terms of language, studies published in English and Spanish were considered for inclusion. Articles were excluded if the health programmes had not been implemented or if they did not determine the characteristics of the programmes or the results obtained. Studies whose publication date exceeded 10 years were also excluded.

### 2.4. Article Selection and Data Extraction

In an initial identification phase, articles were obtained from the following databases: PubMed (*n* = 833), Cochrane (*n* = 864), EMBASE (*n* = 1119), CINAHL (*n* = 445), Web of Science (*n* = 339), and SCOPUS (*n* = 653), resulting in a total of 4253 records. During the screening phase, duplicate studies (*n* = 1358) and those written in a language other than English or Spanish (*n* = 44) were manually removed. Subsequently, articles were excluded if they were not related to the topic in question according to their title and/or abstract (*n* = 2167). In the eligibility phase, two reviewers from the research team independently assessed the 684 articles. Those that did not enable full-text access (*n* = 13) or whose publication date was more than 10 years old (*n* = 562) were excluded, followed by a further 98 articles that did not meet the eligibility criteria. Ultimately, a total of 11 articles were included. This process of article selection is reflected in the flowchart (Figure 1). 

### 2.5. Data Extraction and Synthesis

Two researchers separately extracted the data from the studies included in the review. The methodology proposed by JBI [30] was followed, taking into account the objectives and research questions of the review. A data extraction table was created with the following headings: author, year and country, type of design, study objective, population, main health condition of care recipient, intervention, duration of study and follow-up, study variables and assessment tools, and finally, main results. The method followed for data extraction was tested with one of the articles being selected for inclusion in this scoping review. The reviewers did not find any discrepancies in the data extracted nor encounter any problems in the extraction process. Therefore, the two reviewers used this method to extract data from all of the records. Lastly, a third peer reviewer confirmed the accuracy and integrity of the previously extracted data. A summary of the main data extracted can be found in Table 1. 

### 2.6. Quality Assessment

Quality assessment was conducted using the JBI Critical Appraisal of Evidence Synthesis tools. Two reviewers independently performed quality assessment of the studies, which were then checked by the most experienced researcher included in the study. For this purpose, two tables were created, in which the selected articles were included according to their design. One table was designed to assess the quality of randomised controlled trials (RCTs) and another for quasi-experimental studies. The columns of each table correspond to the items to be assessed in accordance with the respective tools for each type of article. A quality score was then obtained for each study.

## 3. Results

### 3.1. Characteristics and Samples of the Study

The articles included in this review (*n* = 11) involved a total of 1020 older adults who undertake the role of primary family caregiver for their partner. The vast majority of these caregivers are women, as can be seen in the table of results (see Table 1).

The 11 articles included were published between 2015 and 2022. The studies were conducted in different countries: USA (*n* = 3), Sweden (*n* = 2), Finland (*n* = 1), France (*n* = 1), Israel (*n* = 1), Netherlands (*n* = 1), and China (*n* = 1). One of the articles does not specify the country where the study was implemented but does reflect that it was conducted in Central Europe and South Asia. Six of the articles are randomised controlled trials (RCTs), and one article is a quasi-experimental study. The rest are pilot studies (*n* = 4) that can be classified as randomised (*n* = 2) and non-randomised (*n* = 2).

### 3.2. Programmes Implemented and Evaluated to Improve Health

Several programmes were designed to improve the health of older adults who undertake the role of primary family caregiver for their partner. The interventions implemented were carried out by psychologists specialised in geriatrics and gerontology (*n* = 6), as well as by multidisciplinary teams (*n* = 5).

Monin et al. [26] implemented the ‘WOOP’ programme, where caregivers completed daily WOOP cards (Wish, Outcome, Obstacle, and Plan) for 16 days, supported by phone calls every 3 days for monitoring and emotional support. The control group received fewer follow-up calls.

The study conducted by Wawrziczny et al. [9] implemented a 7-week intervention with 90 min sessions covering topics like coping and support networks. Prior to the sessions, a therapist conducted an initial semi-structured interview to identify the needs of the caregivers. Caregivers were given workbooks and received tailored support based on an initial interview. The control group met with a nurse or social worker for standard support.

Milbury et al. [25] carried out a couple-based (caregiver-patient) Tibetan yoga programme (CTYP). The programme consisted of five components: (1) deep breathing awareness with visualisation, (2) breath-holding exercises, (3) mindfulness and focused attention through guided meditation, (4) Tsa Lung movements, and (5) a brief compassion-based meditation. The CTYP programme focused on promoting spiritual well-being in older adults who take on the role of primary family caregiver for their partner with the aim of reducing the burden they experience as they watch their partners undergo a difficult course of treatment.

Kim et al. [7] implemented a music-based intervention in which caregivers were provided with nostalgic Swedish music designed to elicit positive emotions. The caregivers had the flexibility to listen to the music at their convenience, while a coordinator conducted weekly check-ins to ensure smooth participation.

Laakkonen et al. [32] implemented a psychosocial rehabilitation programme aimed at enhancing self-management and empowerment for caregivers. The programme began with home visits to assess caregiver’s preferences and needs, followed by group sessions focused on sharing experiences, promoting empowerment and developing problem-solving skills and resource utilisation.

In the study by Shah et al. [33], telephone support groups were conducted to reduce caregiver burden. The groups were led by a nurse specialised in caregiver training and supported by a social worker and a nurse, both specialised in the care of Parkinson’s patients. Group sessions were held once a week for 90 min via teleconferencing. The topics to be discussed were included in a guide and focused on caregiving education, skills training, problem solving, and support. 

The study by van Knippenberg et al. [27] had three arms, with two intervention groups participating in the so-called ecological momentary intervention (EMI)-based programme ‘Partner in Sight’, which consisted of self-management monitoring using the experience sampling method (ESM). The first intervention group received three face-to-face feedback sessions with a psychologist. The second intervention group, referred to as the pseudo-intervention group, did not receive feedback using the ESM. Alternatively, a semi-structured interview about the participant’s well-being was conducted. By contrast, the control group received usual care. 

Werner et al. [10] implemented the NYUCI/Lituf programme to improve caregivers’ coping skills through tailored sessions and family counselling. In addition, participation in local support groups and ad hoc counselling was encouraged. The control group received usual care, which was not described in greater detail by the authors. 

In the study by Chen et al. [5], the intervention was conducted using reminiscence therapy based on the theory of positive psychology (RTBPPT). The RTBPPT was divided into three sections, focusing on memories, positive caregiving traits, and social support provided to the family caregiver. 

Pandya’s study [8] implemented two different programmes simultaneously for older adult caregivers of their spouses. The first one consisted of online meditation guided by two meditation experts. The second programme involved entertainment activities, such as playing board games, listening to music, or watching documentaries. 

Lastly, the study by Ågren et al. [34] conducted an intervention based on psychoeducational support that started from the second week after hospital discharge. This intervention consisted of three sessions (one face-to-face session and two sessions by telephone calls). Couples facing difficulties in managing their psychosocial distress could be referred to a social worker. Meanwhile, the control group received the usual care provided in these cases. 

### 3.3. Characteristics (Duration and Frequency) of the Programmes

The programmes reviewed ranged in duration from 5 weeks to 2 years. Firstly, the programmes implemented by Pandya [8] and Ågren et al. [34] ran for one year, while the programmes developed by Werner et al. [10] and Laakkonen et al. [32] had a duration of 2 years. Laakkonen et al. [32] conducted assessments at three points in time: at baseline, at 3 months, and at 9 months, concluding the intervention at 24 months. Programmes implemented by Monin et al. [26] (3 months), Wawrziczny et al. [9] (10 weeks), and Milbury et al. [25] (5–6 weeks) ran for shorter periods and included weekly sessions in some cases. Several studies, such as those by Kim et al. [7], Shah et al. [33], and Knippenberg et al. [27], had a total duration of 8 weeks, with follow-ups occurring through calls or teleconferences. 

### 3.4. Health Conditions of the Informal Care Recipients

The majority of the articles included in this review focused on older adults assuming the role of primary family caregiver for their partner with dementia (*n* = 6) [7,9,10,26,27,32]. Two of the studies focused on older adults undertaking the role of primary family caregiver for their partner with cancer [5,25]. Shah et al. [33] focused their research on older adults who were the primary family caregiver for their partner with Parkinson’s disease, while the study by Ågren et al. [34] addressed older adults taking on the role of primary family caregiver for their partner with post-operative heart failure. Finally, one of the studies focused on older adults as the primary family caregiver for their partner with acquired late-life disability, unrelated to any particular health condition [8].

### 3.5. Intervention Outcome Domains

For an older adult, taking on the role of primary family caregiver for their partner is a risk factor that impairs their physical, social and, most importantly, mental and emotional health [8]. Addressing the care of older adults who take on this role is essential for preserving the balance in community health, as caregiver burden leads to poor care for the care recipient, which in turn increases the cost of services and hospitalisations [32]. Though the health programmes implemented in the studies included in this review all focus on the psychological health of the caregiver [5,7,8,9,10,25,26,27,32,33,34], the interventions can be grouped into different domains: stress, depression, caregiver burden, and caregiving skills.

### 3.6. Evaluated Results

#### 3.6.1. Stress

The level of stress perceived by the family caregiver was assessed in three studies [7,26,27]. Kim et al. [7] used the Perceived Stress Scale (PSS-13) before and after the intervention (at 8 weeks), finding no significant improvement in the total scale score (*p* = 0.109) but a significant improvement in the coping subscale score compared with the control group (*p* = 0.048). In the study by Monin et al. [26], the 10-item Perceived Stress Scale was used to assess the level of perceived stress and found a significant decrease in the perceived stress levels after the intervention (at 3 months) (*p* = 0.014). In the study by van Knippenberg et al. [27], the Perceived Stress Scale was also used, obtaining significant differences in the level of stress at two months (*p* = 0.004).

#### 3.6.2. Depression

Depression was assessed in seven of the eleven studies [7,9,10,25,26,27,33]. Kim et al. [7], using the Patient Health Questionnaire-9 (PHQ-9), found no significant differences in depression levels between groups after 8 weeks of music intervention (*p* = 0.106). Milbury et al. [25], using the Center for Epidemiologic Studies Depression Scale (CES-D), observed decreased depressive symptoms with a 6-week Tibetan yoga programme, but without significant within-group improvements (*p* = 0.51). Changes in depressive symptomatology were also not observed in the ‘Partner in Sight’ programme (*p* = 0.137) implemented and evaluated by van Knippenberg et al. [27]. Monin et al. [26], also using the CES-D scale, reported a significant decrease in depressive symptoms during the 3-month intervention (*p* = 0.04), but these differences were not statistically significant when compared with the control group [26]. Shah et al. [33] found a decrease in mean depression scores using the Geriatric Depression Scale (GDS) after two months of tele-support, but the results were not statistically significant (*p* = 0.57). Werner et al. [10], also using the GDS, observed significantly lower depressive symptoms in the intervention group after two years (*p* = 0.0329), though no differences were found during follow-ups home visits at 4 months. Lastly, the study by Wawrziczny et al. [9], which used the Hospital Anxiety and Depression Scale (HADS) to measure depression, found no significant differences between groups after 10 weeks of intervention (*p* = 0.84).

#### 3.6.3. Quality of Life

Three studies assessed the health-related quality of life (HQL) of older adults who take on the role of primary family caregiver for their partner [25,26,32]. Laakkonen et al. [32] used the RAND-36 scale (mental and physical component) at baseline, at 3 months, and at 9 months and found no significant differences in the participants’ quality of life for any of the components assessed (i.e., physical and mental) at any assessment point (i.e., 3 months, 9 months, and 2 years). Milbury et al. [25], using the SF-36 tool, also found no significant improvements 6 weeks after the intervention. The study by Monin et al. [26] measured quality of life using the Quality of Life in Alzheimer’s Disease (QOL-AD) scale. The quality of life of the caregivers who participated in the WOOP significantly improved compared with the control group (*p* < 0.001). 

#### 3.6.4. Caregiver Burden

Family caregiver burden was assessed in four studies [5,8,33,34]. In the study by Ågren et al. [34], it was assessed using the Perceived Caregiver Burden Scale (CBS), finding no significant decrease in burden after one year of psychoeducational sessions. Two studies [5,33] used the Zarit Burden Interview (ZBI). In the study by Chen et al. [5], significant differences were found between the groups in favour of the group that participated in the positive psychology therapy-based programme (*p* < 0.001). In the study by Shah et al. [33], significant improvements in this variable were also found in the intervention group after participating in a tele-support programme (*p* = 0.003). In Pandya’s study [8], assessed by means of the Burden Scale for Family Caregivers (BSFC-s) tool, significant differences were found after the intervention between the intervention group and the control group (*p* = 0.001).

#### 3.6.5. Other Results

Pandya’s work [8] assessed the mental health of older adults who undertake the role of primary family caregiver for their partner using the Mental Health Inventory (MHI-38). The study found significantly higher scores for psychological well-being and lower scores for psychological distress (*p* = 0.001) amongst those who participated in the Internet-based meditating programme when compared with the control group. 

Two studies assessed the family caregiver’s perception of their work and competence as a caregiver [9,27] using the Short Sense of Competence Questionnaire (SSCQ). Both studies found significant differences in the perception of being well prepared and having the necessary competencies to assume the caregiving role, in favour of the intervention group (*p* < 0.001).

Anxiety was assessed in two studies [25,27]. Milbury et al. [25] used the Brief Symptom Inventory-18 (BSI-18) questionnaire and found a significant reduction in anxiety after 6 weeks of the CTYP programme (*p* = 0.04). van Knippenberg et al. [27] assessed anxiety through the anxiety subscale of the Hospital Anxiety and Depression Scale (HADS-A) and found no significant differences between the groups (*p* = 0.226). This same study also assessed positive and negative feelings toward care using a four-item Likert scale for positive affect and a seven-item Likert scale for negative affect, ranging from one ‘not at all’ to seven ‘very much’. Lower levels of negative emotions were found in the intervention group when compared with the pseudo-intervention group (*p* = 0.007) and the control group (*p* = 0.001).

Laakkonen et al. [32] analysed the use and cost of health and social services over two years, finding no increase in service usage or costs following the self-management rehabilitation programme.

### 3.7. Quality Assessment

For the quality assessment of the different articles, the JBI tools were used for each of the included studies: randomised clinical trials (RCTs) [35] and quasi-experimental studies [36]. The quality assessment tool for RCTs [35] was used in eight of the included studies [5,7,8,10,26,27,32,34] (see Table 2). The RCT scores averaged 8–10 out of a total of 13 items, indicating average to good quality. 

All RCTs complied with the randomisation process in allocating patients, measuring reliable outcomes and using appropriate statistical analyses. These aspects strengthen the overall reliability of their findings. However, most RCTs showed important limitation, including unclear regarding allocation concealment in six studies [5,7,10,26,27,34], lack of blinding of administrators in seven studies [5,7,8,10,26,32,34], and lack of blinding of outcome assessors in six studies [7,8,10,26,32,34].

The quality assessment tool for quasi-experimental studies [36] was used in three of the included studies [9,25,33] (see Table 3). The score for the quasi-experimental studies was 4–8 out of a total of 9 points, reflecting variable quality. All three studies completed follow-up and reliably measured the different outcomes obtained, but only one of the studies had a control group and thus a comparison with the intervention group [9]. In addition, two of the quasi-experimental studies were unclear about the outcome measurement tools in both groups [25,33]. 

In conclusion, the overall quality of the studies included in this review was acceptable. The quality assessment revealed that the RCTs generally demonstrated moderate-to-good quality, despite limitations in allocation concealment and blinding that may introduce bias. The quasi-experimental studies showed greater variability, with the absence of control groups and unclear outcome measurements being critical limitations in two studies. These issues should be considered when interpreting the overall findings.

## 4. Discussion

The first specific objective of this scoping review was to identify the health programmes designed for older adults who are the primary family caregivers for their partners and assess their evaluation outcomes. Our findings in this regard show that the health programmes implemented to support older adult family caregivers of their partners are widely varied. Some programmes focused on emotional support and psychosocial rehabilitation, which aligns with the growing body of evidence emphasising the importance of emotional and psychological support in reducing caregiver stress [37]. The findings from other studies implementing these types of interventions suggest that emotionally supported and empowered caregivers are not only more actively involved and feel more competent in their caregiving roles but also show an improved overall well-being [38,39]. Other included studies implemented a tailored intervention with initial assessments to identify caregivers’ needs, aligning with authors who advocate for person-centred care that acknowledges the unique circumstances of each caregiver [40]. Similarly, other studies included in this scoping review implemented complementary interventions such as couple-based Tibetan yoga or music therapy. These programmes not only address physical health but also promote emotional and spiritual well-being [41], which is crucial for older adults to manage the stress and burden associated with being the primary family caregiver for their partner [42]. In addition, the telephone support groups and the real-time feedback implemented in some studies could be particularly relevant in today’s context, reinforcing the need for adaptive and responsive strategies in programmes aimed at improving health outcomes amongst older adult who are the primary family caregiver for the partners [43].

The second specific objective of this scoping review was to describe the key characteristics of the health programmes, including their duration and frequency. Research indicates that longer interventions can lead to more sustained improvements in caregiver well-being [44]. Concurring with these findings, our scoping review shows that the health programmes that ran for at least one year allowed for deeper engagement and ongoing support, potentially leading to more significant reductions in caregiver burden and stress amongst older adults who assume the primary family caregiver role for their partners. This is consistent with previous findings, which suggest that extended interventions provide caregivers with the necessary time to adapt to new coping strategies and integrate them into their daily routines [45]. Moreover, the frequency of sessions within the programmes included in this scoping review could have also influenced their effectiveness [46]. Programmes that incorporate regular check-ins can enhance caregiver engagement and provide continuous support, which is essential for managing the dynamic challenges of caregiving [47], which could contribute to explaining why some of the studies in this scoping review have been more effective than others at improving the targeted health outcomes amongst older adults who assume the primary family caregiver role for their partners.

The third and fourth specific objectives of this scoping review were to identify the main health condition of the people receiving informal care and to explore the outcome domains addressed by the interventions. A substantial body of literature indicates that the type of health condition being managed—such as dementia, cancer, or Parkinson’s disease—affects the caregiver’s experience and the specific support they require [48]. For instance, many of the included studies focus on older adults caregiving for individuals with dementia, which often involves managing complex behavioural symptoms and cognitive decline that can lead to increased caregiver stress and burden [49,50]. This could have influenced the overall findings of this scoping review. Moreover, research suggests that the outcome domains targeted by health interventions play a crucial role in determining programme effectiveness. Programmes that address multiple domains simultaneously tend to yield better outcomes in family caregivers [51,52], which has been corroborated by our findings.

The last specific objective of this scoping review was to evaluate the outcomes that had been measured in the included studies. The main outcomes assessed in these studies were perceived stress, depression, anxiety, quality of life, and caregiver burden. Regarding perceived stress, our scoping review found differences across the studies that may be attributed to the nature of the interventions. For example, the music-based approach may provide emotional relief but not directly reduce stress levels as effectively as the structured goal setting involved in the WOOP strategy. Additionally, using real-time feedback may have allowed caregivers to adjust their coping strategies dynamically, leading to more immediate stress reduction [53]. Regarding depression, anxiety, and mental health outcomes, our scoping review shows that not all interventions improve older adult family caregivers’ psychological well-being. The differences between studies may be attributed to the nature and duration of the interventions; shorter interventions may provide immediate emotional relief but lack the depth needed for sustained changes in anxiety or depression levels [54,55]. Additionally, the specific measures used to assess mental health outcomes may also contribute to these differences, as varying scales can yield different insights into caregiver experiences [56]. Regarding quality of life, our scoping review shows that various interventions (i.e., Tibetan yoga, WOOP, and self-management groups) can significantly impact the quality of life of family caregivers. These findings align with the existing literature that underscores the significance of tailored interventions in enhancing caregivers’ quality of life [57]. The similarities in findings across these studies may be attributed to the shared focus on holistic and person-centred approaches that address the unique challenges faced by caregivers [58]. In terms of caregiver burden, our scoping review shows that reminiscence therapy based on positive psychology, an Internet-based meditation programme, a psycho-educational intervention, and a tele-support group could reduce older adults family caregivers’ burden. Nevertheless, although the tele-support intervention provided valuable resources, the overall reduction in caregiver burden was not as pronounced as in the other three studies. This discrepancy may be attributed to the nature of the interventions; while the first three interventions focus on enhancing emotional well-being, tele-support may primarily provide informational support without directly addressing emotional needs amongst older adults who are the primary family caregiver for their partners [59,60]. 

The findings from this scoping review highlight the critical need for tailored health programmes that address the unique challenges faced by older adults who assume the role of primary family caregiver for their partners. Interventions should not only focus on immediate psychological relief but also on equipping caregivers with coping strategies and practical skills to manage their caregiving roles effectively [61,62]. Additionally, the integration of technology in delivering interventions, such as Internet-based support programmes, may enhance accessibility and engagement among caregivers [63,64]. Future research should explore the long-term effects of these interventions and investigate the mechanisms through which they exert their effects. Understanding the specific components that contribute to successful outcomes will be essential for refining existing programmes and developing new interventions that meet the diverse needs of caregivers [65,66]. Furthermore, studies should consider the cultural and contextual factors that influence caregiving experiences, as these can significantly impact the effectiveness of interventions [67,68].

This scoping review has several limitations. We excluded studies that did not specify whether the mean age of participants was over 65 years or whether they assumed the role of primary family caregiver for their partner, even though they implemented health programmes for caregivers. It may be that some of these studies would have met these criteria, and it would have been interesting to analyse their results. Moreover, it is worth noting the heterogeneity of the studies in terms of the healthcare professionals who carried out the programmes. It is also important to consider the heterogeneity of the programmes themselves, which makes it difficult to ensure that the various programmes included in the study would be effective in other contexts. Similarly, the inclusion of studies that are 10 years old may detract from the relevance of the interventions evaluated in the current global context. The quality of the articles included may limit the validity of the findings of this review. Lastly, one of the main limitations of this review is the lack of studies specifically addressing the role of women as primary family caregivers for their partner, even though the majority of the participants were women. Similarly, although the caregivers participating in the studies were aged 18 years or older, the majority of participants were over the age of 65 years, highlighting the need for further research on this age group, as they are often under-represented in the literature and are identified merely as ‘caregivers’. Given that women are largely responsible for caregiving, future studies should focus on their experiences and the particular challenges they face as older women who take on the role of primary family caregiver for their partner, which would allow for more targeted and effective interventions to be designed for this population.

## 5. Conclusions

This scoping review aimed to examine and map health programmes that have been implemented and evaluated with the aim of improving the health of older adults who assume the role of primary family caregiver for their partner. The results highlighted significant variability in the effectiveness and structure of the interventions. While some interventions, such as positive psychology therapy and guided meditation, have shown significant benefits in reducing caregiver burden and psychological distress, others have not had a clear impact on improving quality of life or reducing depressive symptoms. In addition, longer interventions seem to have a greater potential for yielding lasting improvements, suggesting that sustained support is crucial. On the other hand, the lack of significant improvements in certain studies underscores the necessity for tailored interventions that address the specific emotional, physical, and caregiving challenges faced by individual caregivers. Additionally, the results highlight the need for accessible and well-structured programmes that take into account the diverse needs of caregivers. This review also emphasises the importance of ongoing efforts to develop programmes that are adaptable and sensitive to the unique circumstances of each caregiver. Despite the heterogeneity of the programmes and the variability in the methodological quality of the studies, the findings underscore the urgent need for personalised and accessible interventions to improve the health of older adults who assume the role of primary family caregiver for their partner.

## Figures and Tables

**Figure 1 healthcare-12-02523-f001:**
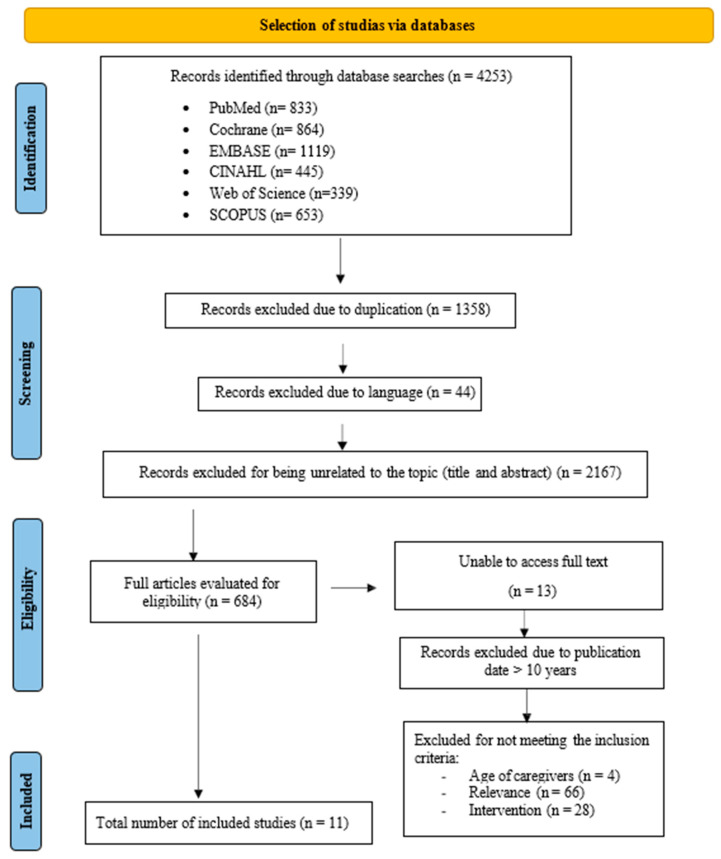
Flowchart of the study selection process.

**Table 1 healthcare-12-02523-t001:** Research articles included in this scoping review.

Author, (Year) and Country	Type of Design	Study Population	Study Objective	Main Health Condition of Care Recipient	Intervention	Duration of Study and Follow-Up	Study Variables and Assessment Tools	Main Results
Monin et al. (2022) [26] USA	RCT	N = 90 (45 couples) Caregivers: IG (WOOP) (*n* = 24): 71% women CG (*n* = 21): 67% women	To examine the viability and effectiveness of the ‘Wish, Outcome, Obstacle, Plan’ method	Dementia	Initial surveys CG: - Educational discussion at initial home visit - 4 follow-up calls - After the study period (3 months) WOOP-guided training (1H) IG (WOOP): - Discussion and handout + guided WOOP training (1H) - 4 follow-up calls. - 16 follow-up WOOP cards to be completed (one a day). - Call regarding the WOOP card every 3 days for 2 weeks	Duration: 3 months - Initial home visit - Follow-up for 16 days - Evaluation after the intervention	Perceived stress: 10-item Perceived Stress Scale. Depressive symptoms: CES-D Quality of life: QOL-AD Positive and Negative Affect: Positive and Negative Affect Scale	After the intervention: - Decrease in perceived stress (*p* = 0.014) - Significant increase in quality of life (*p* = 0.001) and positive affect (*p* = 0.003)
Wawrziczny et al. (2019) [9] France	Quasi-experimental	N = 102 IG (*n* = 51): 62.7% women CG (*n* = 51): 51% women	To reduce spousal distress by acting on distress determinants	Dementia	Semi-structured interview to identify needs. 7 weekly sessions at home to address the 21 intervention modules (1 h 30 min) Exercises in the workbook to implement in each session	Duration: 10 weeks Initial evaluation Questionnaires before the intervention (T0), after the 7 weekly sessions, after the intervention (T1).	Caregivers’ perceptions of the person with dementia’s daily functioning: IADL Caregiving self-efficacy and preparedness: The 15-item Self-Efficacy Scale and Preparedness for Caregiving Scale The effect of caregiving on caregiver experience: CRA Quality of partner adjustment: DAS Perceived health: 2 questions based on the SF-36 Spousal care distress: PDI Depression and anxiety: HADS	Significant differences in the feeling of being well prepared (*p* = 0.00) and impact on daily routine (*p* = 0.03)
Milbury et al. (2015) [25] USA	Non-randomised pilot study	N = 20 (10 couples): 64.3% of carers are women	To examine the feasibility and effectiveness of CTYP	Non-small cell lung cancer	CTYP during radiotherapy treatment. Printed materials + CD of the programme were handed out in sessions 1 and 5 for individual practice in the sessions without an instructor	Duration: 5–6 weeks 15 sessions (2–3 sessions per week); each session: 45–60 min. Interview before the study (T1) and during the last week of radiotherapy of the patients (T2). Follow-up throughout the study	Psychological distress: CES-D Anxiety: BSI-18 Sleep disorders: PSQI Fatigue: BFI Health related QOL: SF-36 Physical Health Component (PHC); Mental Health Component (MHC) Spiritual well-being: Functional Assessment of Cancer Therapy Spiritual Well-Being Scale Search for meaning: Finding Meaning in Cancer scale	Significant reduction of fatigue (*p* = 0.03) and anxiety (0.04). Markedly significant reduction in sleep disorders (*p* = 0.08)
Kim et al. (2022) [7] Sweden	Randomised pilot study	N = 35 IG (*n* = 24): 66.7% women CG (*n* = 11): 54.5% women	To examine the effects of stress and coping capacity	Dementia	Educational session on the benefits of music prior to intervention. Music-based intervention anytime, anywhere. Telephone call once per week	Duration: 8 weeks Weekly follow-up via telephone calls Interview before and after the intervention	Perceived stress and coping skills: PSS-13 Depression: PHQ-9 Perceived Health Status: SF-36	Significant improvement in the coping subscale of the PSS-13 in favour of the IG (*p* = 0.048). There was an improvement in the PSS-13 total scores for the IG compared with CG.
Laakkonen et al. (2016) [32] Finland	RCT	N = 136 IG (*n* = 67): 64.2% women CG (*n* = 69): 60.9% women	To compare the effects of group-based self-care rehabilitation with usual care	Dementia	A group psychosocial rehabilitation model focused on the promotion of self-management skills. Follow-up on the use and cost of health and social services	Duration: 24 months Follow-up: baseline evaluation, at 3 months, at 9 months	HQL: RAND-36 (PCS and MSC) Caregiving skills: SSCQ Mastery: PMS Use of health and social services: centralised registries and medical records Cost of services: centralised cost registries	No significant differences in HQL at 9 months (*p* = 0.55). The intervention was carried out without increase in cost. (*p* = 0.51)
Shah et al. (2015) [33] USA	Quasi-experimental pilot study	N = 7 women	To evaluate the feasibility of a telephone support group	Parkinson’s disease	Tele-support group focusing on emotional support and problem-solving skills with the aim of implementing techniques to reduce the burden on the carer	Duration: 8 weeks 1 session per week via teleconference: 90 min	Emotional and physical distress: AMA Caregiver Self-assessment Depression: GDS Caregiver distress: ZBI Caregiver burden and emotional state: Family Caregiver Assessment	No significant differences in any variable. However, for the GDS there was a significant decrease in depressive symptoms. (*p* = 0.057)
van Knippenberg et al. (2018) [27] Netherlands	RCT	N = 76 IG (*n* = 26): 61.5% women Pseudo-intervention group (*n* = 24): 75% women CG (*n* = 26): 65.4% women	To examine the effectiveness of the EMI programme “Partner in Sight”	Dementia	ESM-based intervention for six consecutive weeks, supplemented by three face-to-face feedback sessions with a personal coach (psychologist). Pseudo-intervention group similar to IG without feedback during the sessions. Alternatively, a semi-structured interview on well-being during the preceding 2 weeks	Duration: 2 months Initial evaluation (T0) 6-week intervention Post-intervention evaluation (T1) 2-month follow-up evaluation (T2)	Primary results Sense of competence: SSCQ Mastery: PMS Secondary results Depressive symptoms: CES-D Perception of stress: PSS Anxiety: HADS ESM outcomes: Positive and negative effect	After the intervention: - Significantly higher caregiver competence (*p* = 0.001) - Lower perceived stress (*p* = 0.004) - Non-significant changes in depressive and anxiety symptoms (*p* = 0.13; *p* = 0.22) - Lower levels of negative affect (*p* = 0.001)
Werner et al. (2021) [10] Israel	RCT	N = 100 IG (*n* = 54): 68.52% women CG (*n* = 46): 69.57% women	Compare NYUCI/Lituf with usual care in reducing depression. Compare NYUCI/Lituf with usual care in reducing depression.	Dementia	Two personalised face-to-face sessions with the family caregiver Four sessions with caregiver and family members	Duration: 2 years Follow-up evaluation every 4 months in the 1st year, and every 6 months in the 2nd year by psychologist and social worker	Depressive symptoms: GDS	After the intervention, there were significant improvements in the GDS for the IG compared to the CG. (*p* = 0.026)
Chen et al. (2020) [5] China	RCT	N = 58 IG (*n* = 27): 48.1% women CG (*n* = 29): 34.5% women	To examine the effect of RTBPPT on burden of care, positive feelings and level of hope.	Cancer	Intervention applied through positive psychology. It was divided into three sections: evoking good memories with their partner, positive aspects of being a family caregiver, and their social relationships	Duration: 1 month 8 sessions: 45–60 min	Caregiver burden: ZBI Positive feelings: PAC Level of hope: HHI	Significant differences in the three variables under study between CG and IG in favour of IG (*p* < 0.001) Significant differences within the IG after the intervention in each of the variables: - Caregiver burden (*p* = 0.003) - Positive feelings (*p* < 0.001) Hope: (*p* < 0.001)
Pandya, (2020) [8] Central Europe and South Asia	RCT	N = 162 IG (*n* = 84): 85.71% women CG (*n* = 78): 84.61% mujer	To examine the impact of an online meditation programme on reducing stress and improving well-being. To examine the impact of an online meditation programme on reducing stress and improving well-being.	Acquired late-life disability	Synchronous IMP sessions by former meditation experts + self-practice. Synchronous OLP sessions by geriatric social workers.	Duration: 1 year	Family caregiver burden: BSFC-s Perceived change in care and own well-being: PCI-13 Mental well-being: WEMWBS Mental health (psychological distress and well-being): MHI-38 (PWB y PD)	At one year, IG compared with CG: - Lower family caregiver burden (*p* = 0.001). - Greater perceived change in care and their own well-being (*p* = 0.001). - Increased well-being and decreased psychological distress (*p* = 0.001)
Ågren et al. (2015) [34] Sweden	Randomised pilot study	N = 42 CG (*n* = 17) 94% women IG (*n* = 25) 84% women	To compare psycho-educational support with usual care on the perceived burden on the caregiver	Post-operative heart failure	Post-discharge psychoeducational support provided by a multidisciplinary team.	Duration: 12 months 3 sessions: 30–60 min	Perceived caregiver burden: CBS Care tasks performed: DOBI	No significant differences at 12 months between CG and IG.

Abbreviations (in alphabetical order): BFI: Brief Fatigue Inventory; BSFC-s: Burden Scale for Family Caregivers-short; BSI-18: Brief Symptom Inventory-18; CBS: Caregiver Burden Scale; CES-D: Center for Epidemiological Studies-Depression Scale; CG: control group; CRA: Caregiver Reaction Assessment; CTYP: Couple-based Tibetan Yoga Programme; DAS: Dyadic Adjustment Scale; DOBI: Dutch Objective Burden Inventory. EMI: ecological momentary intervention; ESM: Experience Sampling Method; FACIT-Sp: Functional Assessment of Chronic Illness Therapy—Spiritual Well-Being; GDS: Geriatric Depression Scale; HADS: Hospital Anxiety and Depression Scale; HHI: Herth Hope Index; HQL: Health-related Quality of Life; IADL: Instrumental Activity of Daily Living; IG: intervention group IMP: Internet-based meditation programme; MCS: Mental Component Summary; MHI-38: Mental Health Inventory; NYUCI: New York University Caregiver Intervention; Lituf: Israeli NYUCI programme; OLP: online leisure programme; PAC: Positive Aspects of Care; PCI-13: Perceived Change Index; PCS: Physical Component Summary; PDI: The 14-item Psychological Distress Index; PHQ-9: Patient Health Questionnaire 9; PMS: Pearlin Mastery Scale; PSQI: Pittsburgh Sleep Quality Index; PSS-13: Perceived Stress Scale; PWB: Psychological Wellbeing Scale; QOL-AD: Quality of Life in Alzheimer Disease Scale; QoL: quality of life; RCT: randomised controlled trial; RTBPPT: reminiscence therapy based on positive psychology theory; SF-36: 36-Item Short Form Health Survey; SSCQ: Short Sense of Competence Questionnaire; WEMWBS: Warwick–Edinburgh Mental Wellbeing Scale; WOOP: Wish, Outcome, Obstacle, Plan; ZBI: Zarit Burden Interview.

**Table 2 healthcare-12-02523-t002:** Summary of the critical appraisal of the methodology of the randomised clinical trials included in the review (*n* = 8).

	Randomisation in Participant Allocation	Allocation Concealment	Similarity Between Control Group/Intervention Group	Blinding of Participants	Blinding of Administrators	Blinding of Outcome Assessors	Same Treatment Among Groups	Adequate Description and Analysis of Group Differences	Participant Analysis in Assigned Groups	Same Measurements	Reliable Measurement of Outcomes	Appropriate Statistical Analysis	Appropriate trial Design	Total Quality Score
Monin et al. (2022) [26]	Yes	N/A	Yes	No	No	No	Yes	Yes	Yes	Yes	Yes	Yes	Yes	9/13
Kim et al. (2022) [7]	Yes	N/A	Yes	No	N/A	N/A	No	Yes	Yes	Yes	Yes	Yes	Yes	8/13
Laakkonen et al. (2016) [32]	Yes	No	Yes	No	N/A	N/A	No	Yes	Yes	Yes	Yes	Yes	Yes	8/13
van Knippenberg et al. (2018) [27]	Yes	N/A	Yes	N/A	Yes	Yes	No	Yes	Yes	Yes	Yes	Yes	Yes	10/13
Werner et al. (2021) [10]	Yes	N/A	Yes	N/A	N/A	N/A	Yes	Yes	Yes	Yes	Yes	Yes	Yes	9/13
Chen et al. (2020) [5]	Yes	N/A	Yes	N/A	N/A	Yes	Yes	Yes	Yes	Yes	Yes	Yes	Yes	10/13
Pandya (2020) [8]	Yes	Yes	Yes	Yes	N/A	N/A	No	Yes	Yes	Yes	Yes	Yes	Yes	10/13
Ågren et al. (2015) [34]	Yes	N/A	Yes	N/A	N/A	N/A	Yes	Yes	Yes	Yes	Yes	Yes	NO	8/13

**Table 3 healthcare-12-02523-t003:** Summary of the critical evaluation of the methodology of the quasi-experimental studies included in the review (*n* = 3).

	Difference Between “Cause” and “Effect”.	There Is a Control Group	Compared Participants Were Similar	Similar Treatment/Care	Multiple Pre- and Post-Intervention Outcome Measures	Same Outcome Measurement in Both Groups	Reliable Measurement of Results	Was the Follow-up Completed or Were the Differences Between the Groups Adequately Described and Analysed?	Appropriate Statistical Analysis	Total Quality Score
Wawrziczny et al. (2019) [9]	Yes	Yes	Yes	Yes	No	Yes	Yes	Yes	Yes	8/9
Milbury et al. (2015) [25]	Yes	No	N/A	N/A	No	N/A	Yes	Yes	Yes	4/9
Shah et al. (2015) [33]	Yes	No	N/A	N/A	No	N/A	Yes	Yes	Yes	4/9

## Data Availability

The original contributions presented in the study are included in the articles; further inquiries can be directed to the corresponding author/s.

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
