# Peer review of "Health Programmes for Older Adults Who Are the Primary Family Caregivers for Their Partners: A Scoping Review"

_healthcare, 2024, doi:10.3390/healthcare12242523_

Round 1
Reviewer 1 Report
Comments and Suggestions for Authors
I would like to congratulate the authors on the choice of the study topic and the paper made. It's relevant, well-written and fulfils all the requirements of an article of this nature. Methodologically, it is well explained.
However, some aspects could be improved. On section 3.1 lines 147 to 152 present repeated information, which can be removed.
On table 2, the authors should identify the studies by number, as shown in Table 1. In the study by Monin et al. (2022), there are nine validated categories, but in the final score there are eight. In the line corresponding to van Knip-penberg et al. (2018) there is a yes in Spanish, which should be replaced by yes (in the reliable measure column).
In this section 3.7, the written evaluation should be more detailed and precise in terms of what is considered a good, average or weak study. The authors evaluate the items but do not present a conclusion regarding the quality they set out to evaluate.
The discussion section should have new evidence about the results achieved. The authors only used nine new references in this section. The discussion is too descriptive: study by study. Authors could improve the intersection of the various aspects found.
The conclusion section adds nothing to what already exists in literature. Authors should incorporate and emphasize the findings of this study.
Reviewer 2 Report
Comments and Suggestions for Authors
1. Please retitle it to "Supporting Aging Family Caregiver: A Scoping Review of Health Program."
2. Introduction part: "over 65 years old" is not older adults, but aging or elderly.
3. Line 61-69, please rewrite your primary and secondary objectives in sentences without mentioning the research questions and hypothesis.
4. In the search strategy part, which keywords you used in the first and second phases?
5. The results are lengthy, but the authors categorized them quite well; please rewrite them to be shorter.
6. The discussion is too lengthy; the first paragraph should be erased. The authors re-write the objectives and the results in Lines 548-560; it can be concluded how the interventions are and how they work. The authors should focus on the clinical implications and policy for the aging population's well-being. You introduced most previous studies in the introduction.
7. The conclusion, Lines 592-596, is unnecessary and should be removed; the author must reply only to the primary and secondary objectives.
Reviewer 3 Report
Comments and Suggestions for Authors
Dear, the manuscript is indeed very interesting and well-structured, and represents the object of study adequately. However, there are some structural deficiencies that make it still immature. The abstract is complete and the whole article is well written. The introduction is functional to the objectives of the review, and the sources are appropriate, but the following elements are missing:
1) PRISMA needs to be completely revised because not all the sources cited (PubMed/Medline, Cochrane, CINAHL, Web of Science, Scopus and EMBASE) are inferred from it: the number of publications and the selection process should be inferred for each. PRISMA needs to be completely reorganized, otherwise it appears that the list cited was included only to give the impression that more than one source was consulted but then not actually done so.
2) Missing is 1 schematic table of the strengths and weaknesses of the contents of the introduction and 1 table in the discussions to make it clear what your review consists of, because it is not very clear from reading what your review has innovated in the current literature, and therefore all the work produced would be lost.
3) Conclusions should be better argued.
Based on these considerations, I do not reject the publication for the authors' efforts but suggest major revisions.
Round 2
Reviewer 3 Report
Comments and Suggestions for Authors
The changes made make the manuscript publishable and complete in all its parts.